# Examining the Continuity between Life and Mind: Is There a Continuity between Autopoietic Intentionality and Representationality?

**Wanja Wiese** [1,*] and **Karl J. Friston** [2]

1   Department of Philosophy, Johannes Gutenberg University of Mainz, Jakob-Welder-Weg 18, 55128 Mainz, Germany
2   Wellcome Centre for Human Neuroimaging, Institute of Neurology, Queen Square, London WC1N 3AR, UK; k.friston@ucl.ac.uk
*   Correspondence: wawiese@uni-mainz.de

**Abstract:** A weak version of the life-mind continuity thesis entails that every living system also has a basic mind (with a non-representational form of intentionality). The strong version entails that the same concepts that are sufficient to explain basic minds (with non-representational states) are also central to understanding non-basic minds (with representational states). We argue that recent work on the free energy principle supports the following claims with respect to the life-mind continuity thesis: (i) there is a strong continuity between life and mind; (ii) all living systems can be described as if they had representational states; (iii) the 'as-if representationality' entailed by the free energy principle is central to understanding both basic forms of intentionality and intentionality in non-basic minds. In addition to this, we argue that the free energy principle also renders realism about computation and representation compatible with a strong life-mind continuity thesis (although the free energy principle does not entail computational and representational realism). In particular, we show how representationality proper can be grounded in 'as-if representationality'.

**Keywords:** active inference; autopoiesis; free energy principle; intentionality; life-mind continuity; predictive processing; mental representation

## 1. Introduction

In 1948, Norbert Wiener characterised cybernetics as an approach that studied common principles between living organisms and machines [1]. As such, cybernetics emphasised the continuity between biological and non-biological systems (cf. W. Grey Walter's [2] 'tortoises'; as well as Ross Ashby's [3] homeostat), which is indirectly related to the idea of a continuity between life and mind [4–7]. As is well known, the latter idea is central to various enactive approaches [8], which can be regarded as descendants of cybernetics [9].

Sharing central assumptions with cybernetics [10], the free energy principle (FEP) can be regarded as a framework that also accommodates enactivist assumptions [11–14]. Active inference [15] and other process theories conforming to FEP (such as predictive processing, [16–18]) have been used to model a rich variety of mental phenomena (for recent overviews, see [19,20]). Hence, it does not come as a surprise that FEP has been invoked to argue for a life-mind continuity thesis [21].

Here, we shall investigate to what extent FEP supports a strong life-mind continuity thesis by discussing how concepts provided by FEP can furnish an understanding of various forms of intentionality. When Franz Brentano imported the term "intentionality" to the modern debate, he characterised it as something "we might call, though not wholly unambiguously, reference to a content, direction toward an object [...], or immanent objectivity." [22]. Brentano's characterisation led to a process of trying to clarify the term 'intentionality' and its significance for philosophy of mind. Now, more than a hundred

years later, it is safe to say that the dominant view on intentionality identifies it with *aboutness* [23], or the property of having representational content [24].

Still, some philosophers argue that there is a basic form of intentionality that does not involve representational content. Examples include Evan Thompson's autopoietic notion of intentionality [25], Daniel Hutto's *basic intentionality* [26], or Hutto's and Glenda Satne's *ur-intentionality* [27] (for overviews of enactivist accounts of intentionality and discussion, see [28,29]).

These authors characterise intentionality as a property of entire systems (not of a subject's mental states). For instance, according to Hutto's account, "[b]asic intentionality is not a property of inner 'mental states,' rather it is a directedness of whole organisms to specific worldly offerings" [26]. This directedness (at a goal, or target) is also emphasised by Hutto and Satne [27], who claim that this form of intentionality is "not only conceptually distinct but picks out a quite independent phenomenon from the kind of intentionality that involves semantic content and 'aboutness' [27]". Furthermore, basic forms of intentionality are typically also ascribed to simple organisms, thus emphasising the continuity between life and mind [25].[1]

There are two important questions that suggest themselves here. (i) How strong is the *explanatory* continuity between life and mind [31], i.e., can the concepts and principles that account for a basic form of intentionality also account for higher cognitive phenomena, including phenomena involving representations [7]? (ii) How strong is the *conceptual* continuity between life and mind, i.e., to what extent can a basic form of intentionality be understood as a form of aboutness (despite what most defenders of basic forms of intentionality would claim—but see [32])?

We will argue that FEP supports a strong life-mind continuity thesis because even the most basic forms of intentionality can be regarded as involving a form of 'as if' representationality. More specifically, we argue that FEP supports both the *explanatory* and the *conceptual* continuity thesis. (Note that we will not argue for the even stronger claim that being alive is *necessary* for having a mind. That is, our view is compatible with the possibility of non-living machines that have real mental properties. In particular, the version of the life-mind continuity thesis defended in this paper is compatible with the goals of strong artificial intelligence.)

The rest of this paper is structured as follows. In Section 2 we introduce the free energy principle (FEP) and briefly comment on the recent debate over realist vs. instrumentalist interpretations of FEP. In Section 3 we review how FEP has been used (i) to analyse central concepts of autopoietic enactivism and (ii) to support a strong life-mind continuity thesis. Furthermore, we highlight two challenges for enactive approaches: the problem of the 'cognitive gap' and the 'interface problem'. The former consists in explaining how higher cognitive capacities are grounded in basic mental properties. The interface problem consists in explaining how non-representational and representational processes jointly coordinate action in systems with mental representations.[2] In Section 4 we discuss a recent instrumentalist representationalist interpretation of FEP that—as we argue—suggests solutions to both of the above-mentioned challenges. In Section 5 we explore how the solutions suggested by this account can be extended to *realist* representationalist interpretations of FEP. We argue that a realist account of representation and a strong life-mind continuity thesis can coherently be maintained.

## 2. The Free Energy Principle in a Nutshell

The free energy principle (FEP, [35]) is not a theory of life (although FEP can be constrained and augmented in such a way as to lead to a theory of life, see [36]). Rather, it

---

[1] Note that we do not wish to imply that these notion are equivalent. For instance, Evan Thompson [30] emphasises important differences between phenomenological and enactive notions of intentionality (which typically involve non-representational content) and Hutto's and Myin's notion of basic intentionality, which does not involve content, because Hutto and Myin equate content with correctness conditions.

[2] One strategy to solve these problems consists in showing how the normativity involved in the activity of systems with basic minds can be extended to account for normative sociocultural practices [21,33,34].

is a theory of the existence of self-organising systems [37], or, more generally, a theory of 'thingness' [38]. Not all existing self-organising systems are alive. For instance, FEP also applies to non-biological agents, provided they possess a sufficient degree of independence from their environment. Because of its generality, FEP is suited to clarify which are the properties that living and non-living agents have in common. So, although FEP does not provide an analysis of what it means to be alive, it can shed light on some core properties of living systems [12,21,36,39].

The following explanation of FEP draws on Section 3.2 in [40] (more thorough treatments are presented in [38,41]; see also [42–44]). In the next section, we shall review how the formalism can be connected to research on autopoiesis and enactivism.

FEP starts with the assumption that living systems (and, more generally, all things that exist) have characteristic features that remain invariant (or stable enough to be measured) for a certain period of time. Formally, living systems can be described as random dynamical systems in non-equilibrium steady state (NESS). Since FEP does not entail specific assumptions about the microstructure of living systems, it uses a coarse-grained description in terms of Langevin dynamics [45–47]:

$$\dot{x}(\tau) = f(x, \tau) + \omega. \tag{1}$$

The Langevin equation is a stochastic differential equation, in which the state of the system $x(\tau)$ at time $\tau$ is given by macroscopic variables. These macroscopic variables (with slow dynamics) depend on microscopic variables that change more quickly. The equation does not capture microscopic details. Therefore, it contains both the state-dependent (slow macroscopic) flow $f$ and a (fast microscopic) fluctuation $\omega$, which is a Gaussian stochastic term with mean equal to zero and covariance $2\Gamma$ (so $\Gamma$ is the amplitude of the random fluctuation; see [38] for details). Given initial states of the system, one can describe (future) states of the system only in terms of a probability distribution.

This distribution can change over time. However, this is not the case at non-equilibrium steady state (NESS). A system at non-equilibrium is in exchange with its environment. Furthermore, at steady state, its dynamics are captured by its NESS probability density, which is invariant during the system's existence. A system approaching NESS converges to a random attractor. There are two ways to conceive of this attractor. On the one hand, it can be regarded as a trajectory of systemic states evolving over time, which revisit certain regions of the system's state space over and over again. On the other hand, the random attractor can be regarded as subtending a probability density $p$ over systemic states sampled at random times. The dynamics of this density are captured by what is called the Fokker-Planck equation. When a system is at NESS, the density $p$ is unchanging. The Fokker-Planck equation can be used to derive a lawful relation between the system's flow $f$ and the NESS density $p$. More specifically, it allows us to decompose the system's flow into a dissipative flow (which ascends the gradients established by the logarithm of the NESS density) and a divergence-free, solenoidal flow (which circulates on the corresponding isocontours; see [38] for details). This has particularly interesting consequences for systems that possess a Markov blanket.

The notion of a Markov blanket plays a central role in the formal analysis of living systems provided by the free energy principle. Originally, it was developed by Judea Pearl [48] in the context of Bayesian networks, i.e., directed acyclic graphs. Since then, it has been applied to directed cyclic graphs (i.e., dependency networks) and undirected graphs (i.e., Markov random fields).[3] A Markov blanket of a node $\mu$ in a Bayesian network is defined as a set of nodes $b$ that statistically separate $\mu$ from all other nodes in the following sense: the probability of $\mu$, given states of the blanket $b$ and any other variables $\eta$, is equal to the probabililty of $\mu$, given $b$. A *minimal* Markov blanket is called a 'Markov boundary' but is sometimes also just called 'Markov blanket'.

---

[3]    Some examples of applications of this concept can be found in [49–52].

In the context of FEP, the term 'Markov blanket' acquires a particular meaning. A Markov blanket $b$ is defined as a boundary between a system's internal states $\mu$ and external states $\eta$, such that $\mu$ is conditionally independent of $\eta$, given $b$. Furthermore, the Markov blanket $b = \{s, a\}$ is constituted by sensory states $s$ and active states $a$. There is no direct causal influence of internal states on sensory states and no direct causal influence of external states on active states. Together, internal, external, and blanket states constitute the systemic states $x = \{\eta, s, a, \mu\}$.

Since blanket states are defined in terms of circular causal relations to internal and external states, the notion of a Markov blanket, as used in, for instance [38,41,53], differs from the original construct employed by Pearl. More specifically, FEP uses the notion of a Markov blanket to formally describe dependency relations in a kind of action-perception loop (for discussion, see [54]; see also [55,56]).[4]

This move has further implications. As Bruineberg et al. [57] convincingly argue, the original construct should be conceived as a merely *epistemic* notion: a Markov blanket is a property of a *model*, not of the system that is being modelled. In other words, it only captures a feature of how we represent certain systems. In the context of the FEP, it is assumed that a Markov blanket is a property of the system itself (this is emphasised in [58]). In strong versions of the FEP, the Markov blanket is an existential imperative, in the sense that anything (e.g., a particle or a person) is stipulatively defined by its internal and blanket states. In the absence of a Markov blanket, nothing can be demarcated from anything else. This version of the concept is therefore a *metaphysical* notion (the issue of whether models developed on the basis of FEP should be regarded as mere scientific instruments or as models that are implemented by organisms is also discussed in [42,59]).

Building on a suggestion by Martin Biehl, Bruineberg et al. [57] propose to use the terms 'Pearl blanket' and 'Friston blanket', respectively, for the two different notions. We welcome this suggestion. However, since we will only refer to Markov blankets in the 'Fristonian sense', we shall stick to the term 'Markov blanket', asking the reader to bear in mind that this should be understand as a metaphysical notion. That is, a Markov blanket, as conceived in this paper, is not just a property of some model of a system but the boundary between something and its environment. This is not a metaphysically neutral assumption, but presupposes a realist stance toward organisms, sensory organs, actuators, etc. Moreover, we do not assume here that it is necessary to derive a system's boundary from a mathematical description of its dynamics. However, a formal specification of the system's properties can at least support a (non-deductive) argument to the effect that a system has a certain boundary.

There is more to the critique presented in [57]. For instance, the authors question that Markov blankets—in the 'Pearlian sense'—can be used to demarcate boundaries between agents and their environments. For the purposes of this paper, we shall only assume the following: if the (admittedly strong) metaphysical assumption is made that living organisms have boundaries comprising sensory and active (i.e., blanket) states, then there is a certain conditional independence between internal and external states—as presupposed in, for instance, [38,53].

If internal states are conditionally independent of external states, given blanket states, we can disentangle the system's dynamics by specifying equations of motion for internal states and active states—that only depend on blanket states:

$$f_\alpha(\pi) = (Q_{\alpha\alpha} - \Gamma_{\alpha\alpha})\nabla_\alpha \mathfrak{I}(\pi)$$
$$\alpha = \{a, \mu\} \quad \text{(autonomous states)} \tag{2}$$
$$\pi = \{s, \alpha\} \quad \text{(particular states)}$$

---

[4] More specifically, the blanket is determined in terms of the sparsity structure of the coupling between states. At any scale, blankets are constructed by looking at the (Hessian of the) adjacency matrix and noting which partial derivatives are zero with respect to which other variables, and this forms the basis of the partition.

Here $\mathfrak{I}(x) = -\ln p(x)$ is *self-information* (or surprisal) and $Q$ denotes solenoidal flow. We refer to $\alpha$, comprising internal and active states, as *autonomous states*. Their flow $f_\alpha(\pi)$ only depends on particular (non-external) states $\pi = \{s, \alpha\}$. For this reason, we can describe the dynamics of internal states in terms of a probability distribution over external states—encoded by internal states: for any blanket state $b$, internal states $\mu$ can be mapped to a probability distribution $q_\mu(\eta)$ over external states, given blanket states.

There need not be a unique internal state, given a particular blanket state (for instance, if the space of blanket states has a lower dimensionality than the space of internal states). However, there will be a unique *expected* internal state that can parametrise $q_\mu(\eta)$. Assuming the probability distribution $q_\mu(\eta)$ is sufficiently similar to the actual conditional distribution $p(\eta|\pi)$ over external states (where $p$ is the NESS density), one can express the flow of autonomous states as a functional of $q_\mu(\eta)$. That is, the gradient flow of autonomous states can be expressed as a gradient flow on variational free energy:

$$f_\alpha(\pi) \approx (Q_{\alpha\alpha} - \Gamma_{\alpha\alpha})\nabla_\alpha F(\pi), \text{ with}$$
$$\begin{aligned} F(\pi) &\triangleq E_q[\mathfrak{I}(\eta, \pi)] - H[q_\mu(\eta)] \\ &= \mathfrak{I}(\pi) + D[q_\mu(\eta)||p(\eta|\pi)] \\ &= E_q[\mathfrak{I}(\pi|\eta)] + D[q_\mu(\eta)||p(\eta)] \geq \mathfrak{I}(\pi). \end{aligned}$$

This shows that changes in autonomous states $\alpha$ must, on average, minimise variational free energy $F(\pi)$—which is a functional of the probability density $q_\mu(\eta)$ encoded by internal states. As a consequence of minimising $F(\pi)$ through changes in internal states, $q_\mu(\eta)$ approximates $p(\eta|\pi)$. Hence, expected internal states look as if they perform approximative Bayesian inference.

Because the expected internal states parameterise a probability distribution (i.e., Bayesian belief) the manifold traced out by expected internal states becomes a statistical manifold, variously called as a conditional synchronisation manifold or more simply internal manifold. Crucially, this internal manifold has an information geometry (i.e., one can measure distances between the probability distributions associated with points on the manifold), which we call the *extrinsic* information geometry (because it is the geometry of probabilistic beliefs about *external* states, see [41]). In addition to a manifold given by probability distributions encoded by internal states, there is also a manifold associated with probability distributions *of* internal states. This manifold has an *intrinsic* information geometry that describes the statistical behaviour of internal states (e.g., neuronal dynamics).

The fact that systems that have a Markov blanket—and implicitly are at non-equilibrium steady state[5]—means that they are equipped with a dual-aspect information geometry; namely, an extrinsic geometry pertaining to probabilistic beliefs *about* external states and an intrinsic geometry pertaining to probability distributions *over* internal states. This dual aspect will be central for the discussion in the rest of this paper. For related work on the fundamental distinction between probabilities of states and probabilities represented by states, see [60,61].

## 3. Autopoietic Interpretations of the Free Energy Principle

In [41], we suggested that the existence of a Markov blanket may be central to understanding the relation between mind and body. Here, we shall discuss the ensuing idea of a 'Markovian monism' from a slightly different perspective, and in a bit more detail (for a critical discussion of Markovian monism, see [62]). In particular, we shall determine what FEP and associated blanket-oriented ontologies (a term coined by Jelle Bruineberg) can tell us about the relationships (i) between living systems and basic forms of intentionality (i.e., intentionality involving (goal-)directedness), (ii) and between basic forms of intentionality and representationality (i.e., intentionality involving representational content).

---

[5]  A non-equilibrium steady-state density must exist before a Markov blanket can be identified in terms of the ensuing conditional dependencies.

Determining the relationship between life and mind—from the point of view of FEP—requires identifying core principles of life that can be analysed in terms of concepts provided by FEP. To the extent that this is successful, it is more or less straightforward to evaluate the hypothesis that there is a continuity between life and a basic form of intentionality. The greater challenge consists in determining the relationship between basic intentionality and non-basic intentionality (inovolving representational content).

There are two general interpretations of FEP that are particularly relevant to the discussion at hand: autopoietic enactivist and cognitivist interpretations [21]. Autopoietic enactivist interpretations of FEP claim that FEP supports a strong life-mind continuity thesis, according to which every living system also has a basic mind (with a basic form of intentionality). The main challenge for such interpretations is to show that the same concepts that are sufficient to explain basic forms of intentionality are also central to understanding non-basic intentionality (which invovles representational content).

Cognitivist interpretations typically entail that there is a discontinuity between life and mind: FEP has to be complemented by a process theory (i.e., a theory of how FEP is realised) that posits representational states, in order to understand the mind (for instance, predictive processing is often interpreted as involving representations, see [17,63–67]; for objections against representational views, see [68–72]). Hence, FEP may apply to all living systems, but only some of them have a mind (viz., if they also have representational states, as posited by the relevant associated process theory). This seems to preclude the possibility that all living systems have (basic) minds. We shall argue that an intermediate position, which takes an instrumentalist stance on representations, can support a strong life-mind continuity thesis and can avoid some of the problems that result from autopoietic enactivist accounts.

Note that it is not obvious that the distinction between basic minds (without representations) and non-basic minds (with representations) can be maintained coherently. That is, it could be that either basic minds or non-basic minds do not exist. Tomasz Korbak [73], for instance, argues that there are no basic minds, because the very strategy employed by [74] to account for content in systems with linguistic skills can also be applied to simple systems (such as single cells). Furthermore, if one commits to a fictionalist account of mental representations [70], one can also deny the existence of non-basic minds. Finally, it is possible to maintain a mixed view: drawing on a distinction between mathematical contents and cognitive contents [75], one can be a realist about representations with mathematical contents, but an instrumentalist about representations with cognitive contents. We shall argue that FEP itself supports instrumentalism about representations with mathematical contents. However, if one accepts a mechanistic account of computation [76–79], then realism about representations with mathematical contents follows *for a wide class of systems*.

In principle, this still allows a distinction between basic and non-basic minds, but one according to which basic minds have at most representational states with mathematical contents. Non-basic minds would then be characterized by the fact that they also have (real) representations with cognitive contents, not only representations with mathematical contents. Further scrutiny may reveal that even this weaker distinction—between basic and non-basic minds—must be given up (if one accepts FEP and a mechanistic account of computation), but we shall not argue for this here. We shall only argue for the more general claim that there is a conceptual continuity between basic forms of intentionality (i.e., intentionality in basic minds) and representationality (intentionality in non-basic minds). Even if it should turn out that basic minds don't exist (as argued by [73]), it would still be possible to ascribe two different forms of intentionality to them (i.e., a basic, goal-directed form of intentionality and a higher form of intentionality involving [cognitive] representational content).

### 3.1. Autopoiesis and Sense-Making under the Free Energy Principle

Two central concepts in autopoietic enactivist interpretations of FEP are *autopoiesis* and *sense making*. Autopoiesis is a concept developed by Humberto Maturana and Francisco Varela. Here is a succinct definition:

> An autopoietic system is organized (defined as unity) as a network of processes of production (synthesis and destruction) of components such that these components:
>
> (i)   continuously regenerate and realize the network that produces them, and
> (ii)  constitute the system as a distinguishable unity in the domain in which they exist [80].

Although not obvious, there is a connection between this concept and cybernetic ideas. This becomes more apparent in the original characterisation provided in [4], in which the authors emphasise that "an autopoietic machine is an [sic] homeostatic [...] system which has its own organization [...] as the fundamental variable which it maintains constant." [4]. According to Maturana and Varela, autopoiesis is both necessary and sufficient for life [4]. More recently, it has been argued that the original concept should be complemented by an adaptivity condition [81,82], which requires anticipatory capacities—through which the system is able "to potentially distinguish the different virtual implications of otherwise equally viable paths of encounters with the environment." [82]. In other words, different states of the environment and different options for action acquire a meaning for the system, "a valence which is dual at its basis: attraction or rejection, approach or escape" [83]. States of the world are thus not neutral for such systems but have a meaning—which is captured by the notion of *sense-making* [25].

Autopoietic enactivist interpretations of FEP point out how both autopoiesis and sense-making can be analysed in terms of concepts that are central to FEP. Recall that FEP is a very general, formal way of describing the difference between systems that preserve their integrity when they are faced with external perturbations, and systems that do not. Examples of systems that fall into the second category are candle flames or snowflakes. Such systems "dissipate rapidly in the face of environmental fluctuations (a gust of air or the warmth of the sun)." [11]. Systems that exist over some appreciable timespan, in the sense of having an attracting set, i.e., revisiting the neighbourhood of characteristic states despite external perturbations—of which living systems are a subset—are importantly different. It is worth emphasising that this has two aspects: firstly, living systems are more resilient to external perturbations than candle flames and snowflakes, and, secondly, living systems revisit the same characteristic states over and over again. This corresponds to the two features of autopoietic systems cited above (i.e., *organisational closure* of components that "constitute the system as a distinguishable unity" and *operational closure*, through which the components "continuously regenerate and realize the network that produces them"). Note that the second feature does not entail that living systems are ergodic. It only requires the existence of an attracting set or manifold, on which itinerant trajectories could appear highly non-ergodic. For instance, language is not ergodic but perigraphic [84]. However, living systems are often ergodic with respect to states with simple attractors, for example, vital states such as core temperature or the level of blood oxygenation. Ergodicity involves abstracting away from certain properties of a system and can therefore be regarded as an idealisation that may or may not be apt for describing nonequilibrium steady-states.

Formally, FEP captures both features by the assumption that the system in question has a Markov blanket and that its NESS density has a low volume, (state) space-filling form. Furthermore, since autonomous (internal and active) states can be described as minimising variational free energy, internal activity must approximate Bayesian inference (and be a form of self-evidencing, [64]). This means that internal processes can be regarded as guided by a norm. In other words, the FEP provides a (very general) answer to the question "What *does* a system do, when it stays alive?", as well as to the question "What *should* a system do, in order to stay alive?" [37]. The FEP thereby accounts for a basic form of goal-directedness that is characteristic for basic forms of intentionality [25–27]. (Formally, this is also captured

by the fact that minimising variational free energy entails maximising value, see below and [85]).

Sense-making not only involves an interpretation of a given situation. It also involves a capacity to anticipate and evaluate the consequences of possible action. The extent to which this is possible for a particle or person will depend on the type of generative model it embodies. As Kirchhoff et al. [12] put it: "Sense making can therefore be associated with [...] the capacity to infer the results of future actions given a history of previous engagement with the world, harnessed in the prior probabilities reflected in the generative model" [12].

In summary, if autopoiesis and sense-making are central to understanding basic and non-basic minds, as proponents of the enactive approach believe, then FEP supports a strong life-mind continuity thesis. For FEP provides a formal analysis of properties that are essential to autopoiesis and sense-making and are instantiated by all systems that have a Markov blanket. But how central *are* these concepts to understanding non-basic minds?

*3.2. The Continuity between Life and Mind under Autopoietic Enactivist Interpretations of the Free Energy Principle*

The strong life-mind continuity thesis entails that the very concepts that account for basic minds (i.e., concepts such as autopoiesis and sense-making) are also central to understanding higher minds. Autopoietic enactivist interpretations of FEP [21], in particular, face the challenge of explaining how higher cognition (involving representational content) emerges from basic mental phenomena—this is known as the "cognitive gap" [33].[6]

A related problem, which is not automatically solved by this approach is the 'interface problem' [34,86,87]. This problem arises if one assumes that (i) some motor activities can be explained without reference to (propositional) mental representations in basic minds; (ii) in non-basic minds, the same motor activities are also caused by mentally represented intentions. Hence, there must be an interface between basic and non-basic aspects of the mental system. (The original version of the problem, as described by [86], asks for the interface between non-propositional, sensorimotor representations and propositional representations. This can be generalised to representational and non-representational processes involved in bringing about behaviour.)

Here, we do not want to comment on the prospects of this approach (for discussion, see [34,88]). Instead, we wish to elaborate on an alternative that has so far received less attention (but see [43,89]). The alternative consists in treating states of living systems *as if* they had representational contents. The resulting instrumentalist approach avoids both the anti-representationalism of radical and autopoietic enactivism, and the representational realism of cognitivist interpretations of FEP.

## 4. Representational Instrumentalism and the Continuity between Life and Mind

Maxwell Ramstead et al. [43] argue for an "organism-centred fictionalism" with respect to neural representations. That is, they argue for an instrumentalist (non-realist) position, according to which neural representations are scientifically useful fictions. The authors draw on Frances Egan's account of mental representation [75,90–92] and apply her distinction between mathematical and cognitive contents to the contents of neural representations (see also [93]).

Mathematical contents are determined by computational models (e.g., active inference models), which specify mathematical functions that are computed. Cognitive contents are posited as part of an 'explanatory gloss', i.e., a description that shows how those parts of the system to which a computational model refers achieve a cognitive task (which itself is characterised with reference to cognitive representational contents, see [75]).

Notably, the account offered by Ramstead et al. [43] is organism-centred, in that cognitive contents are not ascribed to neural representations alone, but to internal states of an organism, where the difference between internal and external states is specified by

---

6    One approach extends the normativity entailed by FEP to normative sociocultural practices [21,33,34].

the Markov blanket partition. In other words, the cognitive contents represented by an organism are determined by the organism's phenotype [43]. This dovetails with autopoietic notions of intentionality, which construe intentionality as a property of organisms, not as a property of neural states.

Furthermore, the instrumentalist account entails a conceptual continuity between autopoietic intentionality and representationality: the very properties that are essential for autopoiesis and sense-making (according to the analysis provided by FEP) are also the grounds of the fictionalist ascription of representational content. More specifically, all systems that have a Markov blanket—and are therefore at NESS—maintain a boundary between internal and external states and revisit regions of state space over and over again. As established above, due to the conditional independence between internal and external states, internal states acquire a dual information geometry. This enables one to interpret internal states as representations of external states (given blanket states), i.e., as probabilistic beliefs. These beliefs (and active states) can be expressed as changing in such a way as to minimise variational free energy, which maximises the value of blanket and internal (i.e. particular) states—defined as the log-likelihood of particular states, given a Markov blanket [85]. This accounts both for the organism's sense-making of the environment and licenses the ascription of mathematical contents to internal vehicles (at least in an instrumentalist sense).

Does this approach solve the problem of the cognitive gap and the interface problem? We shall argue that it does, and that its solution is more compelling than solutions provided by autopoietic enactivist accounts. However, if one is a realist about at least some representational states of non-basic minds, then there is a tension with the account presented by [43] (which we try to dissolve in the next section).

Recall that the problem of the "cognitive gap" [33] requires explaining how higher cognition (involving representational content) emerges from basic mental phenomena. An instrumentalist representationalist account of basic minds (i.e., of systems that have a Markov blanket) already provides the resources to account for higher phenomena: internal states can be interpreted as representations with mathematical content. For basic minds, these representations will be relatively simple. Non-basic minds can be modelled using more complex computational models (e.g., models involving *sophisticated active inference*, see [94]). The form of an organism's beliefs—and of the generative model[7] it entails—may be different if it has a non-basic mind, but the fact that it can be described in this way is predicated on the same feature that is essential for the analysis of basic minds, as well: the existence of a dual information geometry.

Similar considerations apply with respect to the interface problem: explaining how the same motor activities can both be caused by mentally represented intentions and non-representational states does not constitute a puzzle. We only have to keep in mind that "non-representational" states of basic minds can be described as if they were representations with *mathematical* contents—whereas representations of intentions and goals involve *cognitive* contents. In non-basic minds, both types of content ascription are grounded in the same computational model, which refers to a single kind of representational vehicle. There is thus no need for an interface between two types of processes (representational and non-representational), because the central difference is a difference in contents: mathematical versus cognitive contents. These two types of content, however, belong to two different types of description of the system. Cognitive contents only figure in the explanatory gloss, which does not purport to be a realist description of the system's internal states.

However, there is a problem if one is a realist about at least some mental representations with cognitive contents. In that case, the interface problem re-arises, and the question how real representations (in non-basic minds) emerge from fictional representations (in

---

7  A generative model is a probability density $\mathfrak{I}(\eta, \pi) = -\ln p(\eta, \pi)$ over external and particular states that suffices to equate the gradients of variational free energy with the gradient flows on self-information $\mathfrak{I}(\pi)$, that can be described as self-evidencing at NESS. See Equation (2). A particle is said to *entail* a generative model in the sense that its dynamics can be described as a gradient flow on a free energy functional of a generative model and a density $q_\mu(\eta)$ that is parameterised by internal states (see [14]).

basic minds) becomes relevant. Of course, it is still possible to rescue the instrumentalist account by adopting solutions to the two problems suggested by enactivist accounts. Here, we shall explore a different strategy, which involves accepting realism, via a realist account of computation.

## 5. Representational Realism and the Continuity between Life and Mind

FEP does not entail realism about representation or computation (by "computation", we always mean physical computation, i.e., computation in concrete physical systems, not in abstract systems such as Turing machines). However, we shall argue that FEP, in conjunction with a process theory such as active inference, supports teleo-based realist accounts of computation [77,79]. Under the assumption that creatures with non-basic minds have real representational states, one can therefore defend a life-mind continuity thesis by showing that representations are grounded in the computations specified by a process theory.

### 5.1. From Minimising Free Energy to Computational Realism

The first step consists in answering the question how computation and representation are related. This is relevant because, as we have seen, a wide class of systems can be described *as if* they performed computations over representations (i.e., physically parameterised probability distributions or Bayesian beliefs[8]), according to FEP. If we want to know under what conditions we are justified in turning this "as if" description into a realist statement about the system, then we should also clarify under what conditions a system really performs a computation. Depending on whether one defends a semantic or a non-semantic account of physical computation, computation can be prior to representation (or not). Hence, determining how "as if" representationality relates to representationality proper will also require determining how computation relates to representation.

Here, we shall presuppose the correctness of mechanistic (non-semantic) accounts of computation [76–79]. Needless to say, that there are other accounts of computation (for an overview, see [95]); furthermore, the very idea that computational properties are intrinsic properties of some physical systems is contested [96,97]. Our goal is not to defend mechanistic accounts of computation, but to show that one can coherently maintain a strong (conceptual) life-mind continuity thesis *and* a realist account of representation, by assuming the mechanistic account.

Mechanistic accounts of computation propose two key requirements on physical computation [95]. Firstly, there has to be a robust mapping between computational states and physical states of a computing system, such that trajectories of physical states correspond to trajectories of computational states. Furthermore, the mapping must also hold under counterfactual circumstances, i.e., if the system were in a different physical state, it would go through a series of physical states that corresponds to the series of computational states specified by the mapping. Secondly, the parts of the physical system referred to by the mapping must have the teleological function to compute.

Unsurprisingly, the claim that some systems have the teleological function to compute is contested, but here we shall not engage in the debate [98]. Furthermore, we shall presuppose the account of teleological function defended in [99]: "A teleological function in an organism is a stable contribution by a trait (or component, activity, property) of organisms belonging to a biological population to an objective goal of those organisms." [99]. Examples of objective goals of an organism are survival and inclusive fitness [99]. Hence, if causal processes in the organism can be interpreted as computations, and if these computations reliably contribute to survival and/or inclusive fitness, then we can say that the organism has the teleological function to compute, according to this account.

Under the FEP, minimising variational free energy is a corollary of survival and—more generally—existence. Survival just is occupying the attracting set that constitutes the

---

8    *Bayesian belief* is read here as a conditional probability distribution.

particle's (or person's) plausible states. The free energy of any particular state decreases with the probability that the particle (or person) in question would be found in that state. Hence, it might seem promising to argue as follows. If processes in an organism can be interpreted as computational processes that minimise variational free energy, then they contribute to an objective goal of the organism—its survival (or, more generally, its existence). Therefore, if states of a system can be described as if they minimised variational free energy, then they can also be described as actual computational states that minimise variational free energy in a realist sense, because they contribute to survival. In particular, it seems that everything that exists for a certain period of time is also a real computational system (a conclusion that comes close to limited pancomputationalism).

However, this argument misconstrues the relationship between FEP and survival. It is not the case that minimising variational free energy has the function of keeping the organism alive. Rather, part of what it means to stay alive is that the system's dynamics can be described as if they minimised variational free energy. Jakob Hohwy [37] provides a particularly lucid description of this point:

> Notice finally that this account is distinct from any *instrumentalist* or *teleosemantic* notion, which would be the idea that systems minimize surprise in order to achieve the (known) goal of continued existence (or surviving, or achieving intermediate goals and rewards); [...]. The FEP account is fundamentally different from such other attempts [...] because the first step in its explanation is to analyse existence in terms of surprise minimization, rather than naturalistically explain one by appeal to the other. [37]

It is thus wrong to say that minimising surprise (by minimising variational free energy) has the teleological function of continued existence, just as it would be wrong to say that natural selection has the teleological function of evolution. For this reason, teleological accounts of computation cannot be applied to free-energy minimisation as such, but have to be applied to processes that *realise* free-energy minimisation. More specifically, we have to turn to process theories such as active inference or predictive coding, which describe specific ways of minimising free energy. For instance, some models involve prediction error minimisation.[9] We can then ask: "What is the function of these concrete physical processes?", and the answer will be: "They have the function of minimising free energy, and they achieve this by minimising prediction error." This avoids the conclusion that every system equipped with a Markov blanket has the teleological function to compute the mathematical functions specified by FEP.

One could object that this position is still too liberal, for it is possible to describe almost every system as an agent performing active inference. For instance, Manuel Baltieri et al. [100] show that the notorious Watt governor can be described as if it implemented a form of active inference (more specifically, the version of active inference presented in [101]). Does this mean that a Watt governor literally minimises variational free energy? Not necessarily; at least it does not follow from the above. We do not claim that every system that can be described using active inference (or predictive processing) has the teleological function to compute the mathematical functions specified by these process theories (although a Watt governor may well be computational system in a more general sense). For the purposes of this paper, we only have to claim that this holds for living systems. A necessary condition for the survival of a living system is that they can be described as if they minimised variational free energy (this follows from FEP); but a Watt governor is not a living system (note that we do not claim that being describable as an agent minimising variational free energy is sufficient for being alive). Hence, it does not follow that a Watt governor literally computes the mathematical functions specified by a process theory of the FEP (although it would be consistent with the mechanistic account of computation to regard the Watt governor as an analogue computing system that has the

---

[9]   In fact, all self-organising systems at NESS can be cast as minimising prediction error, on average. This is because the gradients that subtends the gradient flow or systemic dynamics can always be written as a prediction error, in the form of a Kullback-Leibler divergence.

function to compute the appropriate opening of the valve for maintaining the speed as set—thanks to an anonymous reviewer for pointing this out).

There is another sense in which this position is relatively liberal because it entails that every living organism that can accurately be modelled using a process theory—that accords with FEP—is a computational system. Presumably, this is the case for almost every living organism (but note that this is ultimately an empirical question). However, it does not entail that every living organism computes every mathematical function (so it is not a catastrophic form of pancomputationalism, [95]). Rather, simple organisms will be described accurately using simple generative models, and more complex organisms using more complex generative models. This is what is to be expected if one takes the idea of a continuity between life and mind seriously (without embracing panpsychism).

*5.2. From Computational Realism to Representational Realism*

So far, we have argued that it is at least coherent to be a computational realist under FEP. What about representationalism? Again, we shall not try to counter objections to realist representationalist accounts (as presented, for instance, by [68–72]). Rather, we shall assume that non-basic minds have at least some genuine representations (in line with views defended by, for instance [17,63–67]). The question that is relevant in the context of the life-mind continuity thesis is whether there is a continuity between autopoietic intentionality (involving a fictionalist ascription of representations) and representationality (in a realist sense).

Living organisms literally compute if they can accurately be modelled using a model $m$, under a process theory of how FEP is realised (e.g., active inference). Hence, we can ascribe *mathematical* contents [75] to the vehicles involved in the computational processes specified by the model ($m$). But this is all we need to justify realism about representations with *cognitive* contents as well, under the assumption that $m$ is sufficiently sophisticated (presumably, it must have sufficient parametric and temporal depth[10]) and under the assumption that representationalist interpretations (such as [65]) are correct.[11] An essential part of what grounds these representationalist interpretations is the fact that systems performing predictive processing or active inference implement computations over representations of probability distributions (of external states, given blanket states). But representations with this type of mathematical content already exist in systems with autopoietic intentionality. Hence, there is a conceptual continuity between autopoietic intentionality and representationality: the concepts used to analyse autopoietic intentionality (from the point of view of FEP) are also essential to analysing representations with cognitive contents; furthermore, even systems with mere autopoietic intentionality have (real) representational states—with mathematical contents.

For instance, according to Paweł Gładziejewski's [65] representationalist interpretation of predictive processing, there are structural representations in systems that engage in hierarchical predictive coding, because such systems exploit relations between internal

---

[10] The distinction between parametric and temporal depth speaks to the hierarchical form of generative models entailed by particular dynamics. The parametric depth [102] corresponds to the depth of the hierarchy in which internal states parameterise probability distributions over the parameters of hierarchically subordinate probability distributions. Note that this implies Markov blankets within the internal states, leading to the notion of functional segregation in systems like the brain (e.g., visual cortical hierarchies). Temporal depth may be crucial when accounting for autopoietic intentionality, in the sense that generative models have to cover the consequences of an intended action—that can only be realised in the future. In other words, temporal depth accounts for the dynamics of internal states that look as if they are planning, anticipating and remembering.

[11] Note that there may be a sense in which at least some cognitive contents are 'free lunch' under the free-energy principle: internal states can be regarded as carrying mathematical contents. According to this interpretation, under the normal ecological operating conditions of a free-energy minimizing device, these mathematical contents become semantic contents, or the ontology that the system brings to bear to parse its flow of sensory data. The variational free-energy becomes a measure of how plausible is a specific semantic interpretation of sensory flow. A surviving organism thus has internal representations with mathematical contents that are accurate in representing the 'real world out there' to the extent that they minimise variational free energy in the long run. In this interpretation, even the most deflationary account of contents as mathematical contents leads to an intrinsic semantics, in the sense that the mathematical contents themselves are sufficient to ground representationality given the right environment. This is equivalent to saying that it inhabits certain environments and enters states that are conducive to its continued existence. In other words, its viability conditions are the truth or accuracy conditions of (at least some of its) representations with cognitive contents. We are grateful to Maxwell Ramstead for pointing this out.

representations with mathematical contents (see also [93]). But such types of mathematical content already exist in relatively simple systems, such as single-cell organisms, albeit not necessarily with the parametric depth required to justify the existence of representations with cognitive contents.[12]

*5.3. Revisiting the Interface Problem*

Note that this realist account of computation and representation suggests more or less the same solutions to the problem of the cognitive gap—and the interface problem—as the instrumentalist account. The main difference is the ontological status ascribed to the representations that figure in the explanation of mental phenomena.

In particular, the account suggests the following way of dealing with the interface problem. According to a structuralist representationalist interpretation of predictive processing [65], representations with cognitive contents are structured sets of representations that have mathematical contents (e.g., as in hierarchical predictive coding) and fulfil a representational role. In this case, there is no need for an interface between representations with different types of content, because representations with cognitive contents are constituted by representations with mathematical contents.

One could worry that this does not solve the interface problem if some representations have different formats (e.g., analogue versus digital, see [104,105]); or, if some sets of representations with mathematical contents ground the ascription of representations with cognitive contents in a propositional format, whereas other representations with mathematical contents (in the same system) do not ground any representations with cognitive contents at all. For instance, high-level representations of long-term goals and action policies ground representations with propositional contents, but some low-level representations—that are implicated in predicting the immediate consequences of action—only have mathematical contents. How are representations of long-term goals and intentions translated into sensory predictions that drive action?

In this case, there needs to be an interface indeed. The solution to this interface problem is provided by existing process theories, such as active inference. Pezzulo et al. [106] illustrate this with the example of choosing whether to have dessert at a restaurant (and realising the corresponding actions). The key to the solution rests on using a hierarchical inference scheme with representations of semantic narratives ("I am now at a restaurant and have just finished the main course.") at the top, representations of perceptual and proprioceptive variables at the bottom, and mediating representations (for instance, of affordances) in between. The different levels are related by feedback loops in which, as Pezzulo et al. put it, "goals or prior preferences at one level translate into predictions about sequences of events that provide top–down (empirical) prior constraints on transitions at the level below. In turn, bottom–up messages from lower levels report the evidence for expectations of beliefs generating predictions, thus permitting the higher levels to accumulate evidence (e.g., about progresses towards the goal) to finesse plans." [106]. Furthermore, representations at different levels operate at different timescales (representations of simple and momentary variables at the bottom versus representations of hidden states and goals that are necessary for long-term planning and prediction at the top). The interface between these representations is provided by the processing hierarchy, because "each hierarchical level operates independently and passes the results of its computations to the levels below (and above)." [106].

## 6. Conclusions

The strong life-mind continuity thesis examined in this paper entails that the same concepts that are central to explaining basic minds (which only have a basic form of intentionality) are also central to understanding non-basic minds (which have intentional

---

[12] One could object that it may be necessary to refer to additional features to explain representations with cognitive contents that are not organism-relative, which would suggest a conceptual discontinuity with some types of representation, see [103]—we leave this as an open question for future research.

states with [cognitive] representational content). We have distinguished this *explanatory* reading from a *conceptual* reading of the life-mind continuity thesis: according to the conceptual version of the thesis, it is possible to regard the basic form of intentionality that all living systems possess as a form of representationality.

We have argued that instrumentalist and realist representational accounts are compatible with both the explanatory and the conceptual version of the life-mind continuity thesis. In particular, the conceptual version holds because even basic forms of intentionality involve representations with mathematical contents. Furthermore, we have argued that both types of account suggest elegant solutions to two vexing problems associated with the life-mind continuity thesis: the problem of the cognitive gap and the interface problem. The first consists in showing how explanations of basic minds can be extended to non-basic minds. The interface problem consists in showing how representational processes cooperate with non-representational processes in motor control. The problem disappears for representationalist accounts, if one acknowledges that the alleged 'non-representational' processes involve representations with mathematical (as opposed to cognitive) contents.

We conclude that there is a strong continuity between life and mind, and that there are good reasons for assuming, in particular, a continuity between autopoietic intentionality and representationality (aboutness). We have seen that one can come to this conclusion via different paths (i.e., representational instrumentalism and realism). An interesting project for future research will be to compare and evaluate these paths in more detail.

**Author Contributions:** Conceptualization, W.W. and K.J.F.; formal analysis, W.W. and K.J.F.; writing—original draft preparation, W.W.; writing—review and editing, W.W. and K.J.F. All authors have read and agreed to the published version of the manuscript.

**Funding:** This research was supported by a Wellcome Trust Principal Research Fellowship (KJF; Ref: 088130/Z/09/Z).

**Institutional Review Board Statement:** Not applicable.

**Informed Consent Statement:** Not applicable.

**Data Availability Statement:** Not applicable.

**Acknowledgments:** We are extremely grateful to the audience at the 4th Avant Conference, Porto, at which an early version of this paper was presented, as well as to Maxwell Ramstead for providing highly valuable feedback on a draft of this paper. We also thank the two reviewers of this journal for their extremely helpful comments. W.W. is grateful to Jelle Bruineberg and Joe Dewhurst for correspondence on issues related to ideas presented in this paper.

**Conflicts of Interest:** The authors declare no conflict of interest.

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
