# Peer review of "Examining the Continuity between Life and Mind: Is There a Continuity between Autopoietic Intentionality and Representationality?"

_philosophies, doi:10.3390/philosophies6010018_

Round 1
Reviewer 1 Report
The paper argues that if one accepts FEP and the new mechanistic account of computation, a solution of a problem in the debates over enactivistic approaches to intentionality could be offered. As such, this paper surely fits the Special Issue, although it might be of limited interest of scholars that are not engaged in these debates. The paper is competent and the conclusion is supported by a convincing argument but there are three problems that should be addressed.
(1) Basic minds / non-basic minds
In a somewhat underappreciated paper, Korbak (2015) has shown that the class of biological systems with basic minds is exactly empty. The problem is that, e.g., Hutto and Myin accept that linguistic expressions have contents with satisfaction conditions. But the notion of language, as characterized by Hutto and Myin, applies also to other natural semiotic systems, including cell signaling, which implies that contents cannot be denied to any biological cell on enactivist grounds consistently. Note that the argument presented by Korbak has never been rebutted although Hutto and Myin cited it in their subsequent work (without engaging with its crucial argument, however).
Because of this, it is quite unclear to me whether the problem is real at all. It is not sufficient to say that basic minds have no genuine representations to characterize them; the difficulty is that there might be no such minds at all. Note that Korbak's argument actually leads to the conclusion you seem to reach, albeit in a different manner, i.e., that mechanistic account of computation and FEP jointly imply contents. In other words, it seems that you also deny the existence of basic minds without contents.
(2) Section 2 length and the rest of the presentation
Section 2 seems overly long as compared to the rest of the paper; some of its content has little implication for what is said later. In particular, p. 9, lines 211-222, seem to have no bearing on the discussion later, so these could be shortened. These could be safely removed.
Moreover, the equation (2) (p. 9, no line number), contains symbols that are undefined in the paper, such as I (sorry for the lack of genuine symbols), which is later used in fn 6 on p. 17. These should be properly defined and introduced; otherwise, the equation is cryptic for the reader that has not encountered it before, and those who have encountered it, do not need this introductory section.
There is also some contrast between this section and the rest of the paper in terms of mathematical precision. In section 3.1, a divide between organisms and other NESS systems is introduced by saying that some systems "dissipate rapidly". But some candles burn for days, whereas some mayflies live for 24 hours. There are really a lot of ephemeral biological organisms, and the time span difference does not seem to mark the difference between non-living and living NESS. Moreover, saying that living organisms "revisit the same types of state over and over again" does not mark the proper difference as the candle obviously has an attractor, and also prepares the network of processes to maintain burning etc. Notably, revisiting the same states is not entirely true of human beings with complex linguistic representations, because language is not ergodic but perigraphic (see Dębowski 2020), i.e., on average we do not revisit the same states by uttering the same linguistic expressions. (Obviously, you could idealize this way, and maybe it would be helpful to stress that some of the assumptions of FEP are useful idealizations).
(3) Footnote 10.
Footnote 10 (p. 23) contains, at the end, fairly important claims that viability conditions are satisfaction or truth conditions. They would deserve to be part of the main text and defended against run-of-the-mill radical enactivism argument (Heras-Escribano et. al 2014) that biological normativity has nothing to do with content normativity. Hutto and Myin defend this kind of argument in many places.
Very minor comments:
p. 2, line 36: W. Grey Walter did not use his first name in publications (see the cover of his 'The Living Brain'), and he is usually called 'Grey Walter' or 'W. Grey Walter'. I would add 'Grey' at least. (This is a very minor issue)
p. 22: I don't think mechanists would find ascribing computations to Watt governor even close to pancomputationalism. After all, it's an analog control device whose whole purpose is to measure the speed of the engine and adjust it; this makes it fairly obvious that it has a function of computing (as an analog computer) appropriate opening of the valve for maintaining the speed as set. While there were no defenses of this claim in print, I don't think this would imply any kind of pancomputationalism. Controllers or regulators are obviously artifacts we use, so they have derived functionality, and saying that a class of artifacts is a class of computing mechanisms is not pancomputationalism. Note that limited pancomputationalism claims that all causal structures (or mechanisms) compute at least one function, and catastrophic pancomputationalism says that all physical systems compute all kinds of mathematical function. Only the latter is to be avoided. But your argument come nowhere near.
It would also be entirely unsurprising to find that all biological organisms have some computational machinery in the mechanistic sense; after all, cell signaling mechanisms, immune systems, genetic mechanisms all trade in medium-neutral vehicles, so they should be understood as computational. But of course they do not compute all the functions all the time, just some particular function.
References
Heras-Escribano, M., Noble, J., & de Pinedo, M. (2014). Enactivism, action and normativity: a Wittgensteinian analysis. Adaptive Behavior, 23(1), 20–33. https://doi.org/10.1177/1059712314557364
Korbak, T. (2015). Scaffolded Minds And The Evolution Of Content In Signaling Pathways. Studies in Logic, Grammar and Rhetoric, 41(1), 89–103. Dębowski, Ł. J. (2020). Information theory meets power laws: stochastic processes and language models. Hoboken: Wiley.Author Response
Please see the attachment.

Reviewer 2 Report
The paper presents an interesting argument concerning the relation between life and mind. I won't make line by line comments but rather present two objections that focus on the author's solution to the interface and gap problems, which I find problematic. But first, I would like to make a point that the authors need not respond to, but which would be interesting for them to address, namely the implications of their view for artificial intelligence, if any. Some authors who have written about the relation between life and mind believe that, if there is something essential about the relation between life and mind, then there are negative implications for the prospect of AI, or at the very least, there are implications about why evolutionary approaches to the mind should be preferred (the work of Margaret Boden is relevant here). Is the continuity of life and mind compatible with AI? Readers might be curious.
Now to the two substantial points. The authors rely on Frances Egan's characterization of contents as mathematical and cognitive in order to solve the interface problem (as well as the cognitive gap problem). I shall focus on the interface problem. The first difficulty is that according to some authors, this distinction entails a difference in the format of these representations, or at least, it is possible to interpret this difference as a difference in format (see Fodor, 2007; Maley, 2011, 2018; Montemayor, 2013). If this interpretation is correct, then I cannot see how the interface problem is solved by the authors because analog content is essentially approximate and digital is symbolic or conceptual. A concrete example helps here. All living organisms represent time and have similar clocks. But plants presumably don't represent time the way animals do (it is questionable whether they represent anything at all). And even animals represent time in a more mathematical fashion than humans. The difference between analog (mathematical or map like) and digital formats (propositional or sentence-like) is fundamental because there is loss of information from analog to digital and only digital formats seem to be capable of compositionality and recursion, as well as misrepresentation (see Montemayor, chapter 3 for criteria concerning representation, including misrepresentation, see also Fodor 2007 on the semantic differences between these formats). So if there is a difference in format, then one cannot simply say that the mathematical can be glossed propositionally, because compositionality operates differently depending on the format. More important, there is no clear interface in content because the semantics work radically differently.
Second substantial point: A possible solution to the issue of format that the authors might embrace is following Figdor's proposal for basic aboutness (2020) (or another similar unified account). But if they embrace this account, then it is not clear that they are offering an original solution to the gap or interface problem. Thus, a difficulty emerges: either the author's stated adherence to Egan's distinction entails a difference in format or not. If it entails such a difference, then there is no clear solution to interface problem because analog and digital formats behave differently or have different semantic structure. But if there is no difference at all, then the gap and interface problems dissolve simply by appealing to a unified notion of content and Egan's distinction appears unjustified or unmotivated.
The authors need not get into the problem of semantic content in general, which is a very difficult issue. But a concrete example or argument is needed in order to clarify how exactly they think they solve the formatting problem.
References
Figdor, C. 2020. Shannon + Friston = Content: Intentionality in predictive signaling systems. Synthese.Fodor, J. 2007. Revenge of the Given. In B. P. McLaughlin and J. Cohen (eds), Contemporary Debates in the Philosophy of Mind. New York, NY: Basil Blackwell pp. 105–116.
Montemayor, C. 2013. Minding Time: A Philosophical and Theoretical Approach to the Psychology of Time. Leiden, The Netherlands: Brill.
Maley, C. J. 2011. Analog and digital, continuous and discrete. Philosophical Studies 155(1), 177-131.
Maley, C. J. 2018. Toward analog neural computation. Minds and Machines 28(1), 77-91.
